# Computer Vision for DC Partial Discharge Diagnostics in Traction Battery Systems

Ronan Sangouard [1,2,†], Ivo Freudenberg [1,3,†] and Maximilian Kertel [1,4,*,†]

1 BMW AG, 80809 Munich, Germany
2 Department of Computer Science, Technical University of Munich, 80333 München, Germany
3 Faculty of Electrical Engineering, Darmstadt University of Applied Sciences, 64295 Darmstadt, Germany
4 Department of Statistics, TU Dortmund University, 44227 Dortmund, Germany
* Correspondence: maximilian.kertel@bmw.de
† These authors contributed equally to this work.

**Abstract:** The tendency towards thin insulation layers in traction battery systems presents new challenges regarding insulation quality and service life. Phase-resolved DC partial discharge diagnostics can help to identify defects. Furthermore, different root causes are characterized by different patterns. However, to industrialize the procedure, there is the need for an automatic pattern recognition system. This paper shows how methods from computer vision can be applied to DC partial discharge diagnostics. The derived system is self-learning, needs no tedious manual calibration, and can identify defects within a matter of seconds. Thus, the combination of computer vision and phase-resolved DC partial discharge diagnostics provides an industrializable system for detecting insulation faults and identifying their root causes.

**Keywords:** phase-resolved DC partial discharge diagnostics; computer vision; insulation testing; pattern recognition; automotive industrialization; root cause analysis

## 1. Introduction

Despite the urgency to reduce carbon dioxide ($CO_2$) emissions, the $CO_2$ emissions of the transport sector grew from 2021 to 2022 by 2.1%. The increase would have been even higher without the "accelerating deployment of low-carbon vehicles". Electric vehicles saved 13 million tons of global emissions in 2022 compared to typical diesel or gasoline cars [1]. Hence, the increasing number of battery electric vehicles (BEV) can reduce carbon emissions and improve the sustainable development of the transportation sector. A crucial component of BEVs is the traction battery system, which requires reliable insulation systems to ensure safe and efficient operation. However, the trend towards thin insulation layers in these systems has presented new challenges regarding insulation quality and service life [2]. Thus, a quality test is essential to avoid current and future insulation faults. Future faults are detectable using partial discharge diagnostics. Partial discharges (PD) can be a preprocess of an electrical breakdown of insulation systems [3].

PD diagnostics have been established as a reliable testing method in AC power systems [3]. However, their application in DC power systems, particularly in low- and medium-voltage applications in the automotive sector, is still in the research stage [4,5]. The DC operating voltage defines the requirement for a quality test using DC stress [3]. PDs in the solid insulation system at DC stress are usually at the surfaces of defects due to space charge formations [4,6]. Thus, the aging process defers using an AC or a DC stress. Consequently, DC partial discharge diagnostics should be applied.

In this context, a new approach is transferring AC PD diagnostics via phase-resolved partial discharge (PRPD) patterns to DC systems in low- and medium-voltage applications. The proposed method involves applying a small ripple to the DC voltage and using it as a phase-angle reference for partial discharge diagnostics. This early fault detection [5]

can provide several discharge patterns that refer to faults on and in the insulation system. As well as the AC PRPD patterns [7], the introduced DC patterns should be quantified for establishing an industrial short-term routine test. Until now, DC partial discharge diagnostics have been considered unsuitable for routine tests due to the long testing time of "over tens of minutes or hours" [4], due to a slower recovery time [8]. Considering the thin insulation layers of traction battery systems, the PD rate is higher than in DC high-voltage applications caused by the lower ohmic recharging process of capacitive faults. Additionally, the ripple can accelerate partial discharges in the volume of the defect. Thus, the test method and the application enable a routine test in a short test time. To meet industrial routine test requirements for early fault detection, there is the need for a fully automated system, which is capable of quantifying the PRPD patterns within seconds.

## 2. Related Work

This work investigates the automatic identification of PRPD patterns, which is well studied [9]. However, to the best of our knowledge, the present work is the first to investigate the automatic identification of PRPD patterns of an insulation from a DC power system. The application of DC partial discharge diagnostics provides several advantages:

- Potential root causes can be identified by different patterns.
- Partial discharges in the volume of the defect can be detected.
- Depending on the application, one can simulate the working load of the test object.

In order to apply the testing procedure in a manufacturing environment, an automated pattern recognition system is desirable. It must be capable of identifying the patterns of the test object and thus the potential root causes within seconds. This work provides the primary proof of concept (PoC) for this.

From a machine learning perspective, automated PRPD pattern recognition algorithms can be roughly split into methods based on features identified by experts [10,11] and methods which are purely data-driven [12,13]. In this work, we focus on the latter, which we consider to be more robust towards perturbations of the testing setup and new images. Compared to existing works, we leave the denoising of the images mostly to the pattern recognition algorithms. This avoids tedious calibrations of the noise removal step [14], while further increasing the robustness of the model.

To create partial discharge plots, one needs to decide on their resolution. The optimal resolution of the PRPD patterns for automatic classification has so far been neglected in the literature. We dedicate Section 5.1 to this issue.

Additionally, the automatically learned patterns are checked for plausibility using methods from explainable artificial intelligence (xAI) in Section 5.4. In this way, we involve the existing expert knowledge in the validation procedure.

## 3. Technical Background

AC partial discharge measurements are usually utilized in high-voltage applications. The phase of the AC test voltage is used as a reference to identify defects. The partial discharges (PDs) in faults behave according to the field stress, material properties and geometrical properties. Thus, PDs in different failure routines can be related to phase-resolved partial discharge (PRPD) patterns [4]. This method can also be used in DC PD diagnostics by superimposing a small ripple on the DC stress [15]. Thus, the methodological knowledge can be transferred, supporting the introduction of DC PD diagnostics using PRPDs. The DC PD diagnostic method can be applied using the test setup in Figure 1a. The AC side consists of a controlled AC voltage source, a transformer, a coupling capacitance $C_{ac}$ (1.2 nF), and the PD measurement device Omicron MPD 800 ($PD_{ac}$). It measures PD at the AC side to ensure a PD-free power supply. The diode rectifies the voltage via the transition to the DC side. According to the direction of the diode, positive and negative voltages can be applied. The DC voltage ripple can be set using a smoothing capacitance $C_g$ (20 nF) and a load resistor $R_L$ (5 MΩ). The inductance $L$ decouples the measurement part from the test voltage generation part. The ohmic divider measures the mixed signal at the

test object (TO). The coupling capacitance $C_{dc}$ (1.2 nF) and the MPD 800 ($PD_{dc}$) are utilized to measure the apparent charge and are calibrated to the TO. An electrode (conductive elastomer) [5] connects the test setup to the TO (Figure 1b).

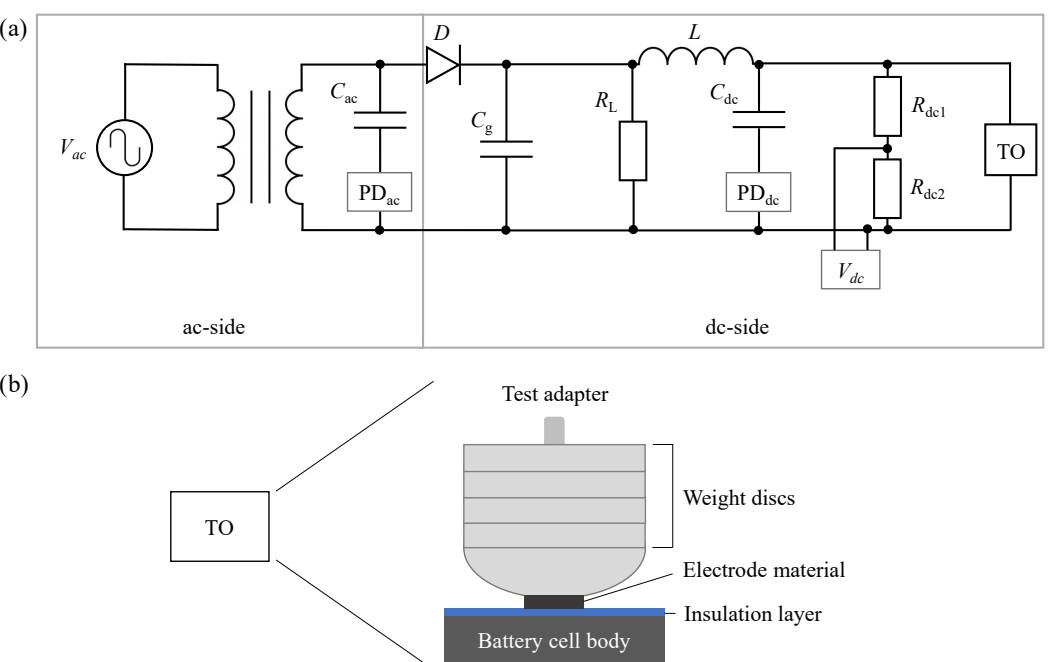

**Figure 1.** Test setup: (**a**) equivalent circuit and (**b**) test object [5].

An integration of the PD current impulses over the frequency domain calculates the discharge magnitude. A bandpass filter is applied according to IEC 60270 [16]. The PDs are related to the phase angle. PRPD patterns are recognizable by overlaying several periods (20 ms) of the ripple. The discrete intensity of discharges is displayed with a color map to identify the PD distribution. Figure 2 shows the usual patterns in thin insulation systems of traction battery systems. The noise with a maximum discharge magnitude of 100 fC ($Q_{IEC}$ = 55 fC) is removed to clearly display the patterns. Figure 2a shows a pylon-shaped pattern representing PDs in the air gap between the electrode and TO. The pylon-shaped pattern is in the range of the maximum absolute voltage. The deviation of the pylon-shaped pattern depends on the distribution of space charge in the air. Solid PDs caused by impurities occur as plateau-shaped PRPD patterns (Figure 2b). The PD behavior of solid PDs can be described by hopping, tunneling and combined processes [17]. The discharge magnitude depends on the local electric field stress and the magnitude of the potential barrier [18] of the specific impurity. Thus, PDs can occur over the total phase range triggered by the field stress of the PD location. Volume PDs form a hill-shaped pattern (Figure 2c), which depends on the gradient of the applied voltage [15]. A quadratic shape and a phase shift, in contrast to the pylon-shaped pattern, characterize the hill-shaped pattern. These patterns are usually measured in a combined pattern, like pylon and plateau, as shown in Figure 2d. In particular, plateau and hill can reduce the life cycle of traction battery systems. Orienting investigations show that these patterns lead to an erosion and a breakdown of insulation. More detailed information about the patterns, effects and physical discharge processes are summarized in [5].

Until now, the patterns have been qualitatively analyzed. The PD rate and the weighted discharge magnitude $Q_{IEC}$ are not suitable for a binary ok or not ok decision in a routine test. Additionally, both parameters cannot provide information about the type of measured defect. However, this information can help optimize the production process by understanding the cause of the defect. To achieve this, an automatic system is necessary to identify and quantify the patterns. Furthermore, the automatic approach should fulfill routine test requirements to optimize cycle times.

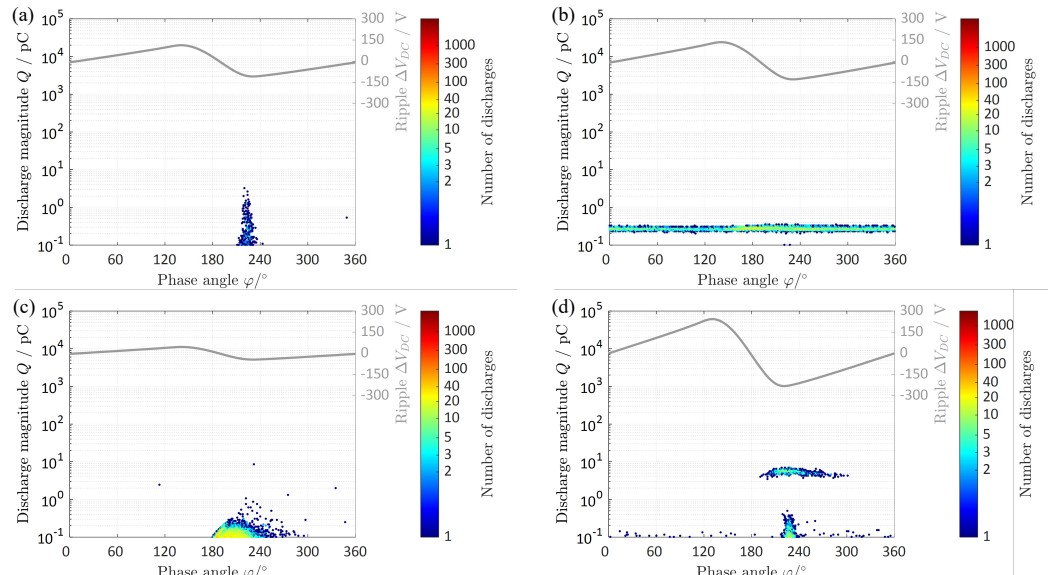

**Figure 2.** Introduced PRPD patterns: (**a**) pylon-shaped pattern at $V_{peak} = -2$ kV; (**b**) plateau-shaped pattern at $V_{peak} = -2.7$ kV; (**c**) hill-shaped pattern at $V_{peak} = -0.85$ kV; (**d**) combined pattern pylon and plateau at $V_{peak} = -4.5$ kV and $t_{test} = 1$ min.

In this work, a full machine learning approach is followed, where patterns are learned autonomously by the system from the PRPD diagrams. As an alternative, one could implement the expert knowledge as a predetermined classification algorithm, e.g., one could count the number of discharges in a certain area of the diagram. However, this kind of approach involves the calibration of many parameters; for our example, there would be the need to specify the following:

- The area;
- A threshold for the number of discharges in that area, above which we assign the diagram to a specific pattern.

This calibration is tedious and the approach is unstable for unseen images or slight variations in the experiment setup. A mixed approach of machine learning and expert systems, as described in [9,11,14], provided unsatisfactory results in a preliminary analysis and was thus not considered further.

## 4. Machine Learning Approach

In this section, the machine learning approach is described. The analysis is based on a dataset of PRPD recordings, which is detailed in Section 4.1. As the procedure aims to identify patterns in PRPD diagrams, their generation is outlined in Section 4.2. This is followed by an explanation of the machine learning algorithms under investigation (Section 4.3) and a description of the methodology (Section 4.4), which we use to compare the algorithms in Section 5 to conclude.

### 4.1. Dataset Description

The data presented in this study are the intellectual property of BMW AG and have been used with their permission. Due to the proprietary nature of the dataset, it cannot be published in its entirety. However, in the following we provide the necessary details to support the reproducibility and validity of our results.

The studied dataset consists of $N = 369$ files. Each file contains the output of one discharge experiment, which consists of a succession of discretized discharges. Inside a file, data are organized in a three-column table. The three columns contain the signed amplitude of the discharge, the phase of the discharge and the absolute time of the discharge,

respectively. Given that the frequency of the discharges and the duration of the experiment vary, the number of rows differs between the files.

An expert assigns a label to each file, based on the corresponding PRPD diagram. Each label is made of a combination of $C = 3$ base classes, namely "Hill", "Plateau" and "Pylon". Figure 3 shows isolated instances of each base class. Table 1 gives the number of files for each class combination. It shows that 28% of the instances contain a "Hill", 74% of the instances contain a "Plateau" and 49% of the instances contain a "Pylon". Thus, the imbalance of the base classes is limited.

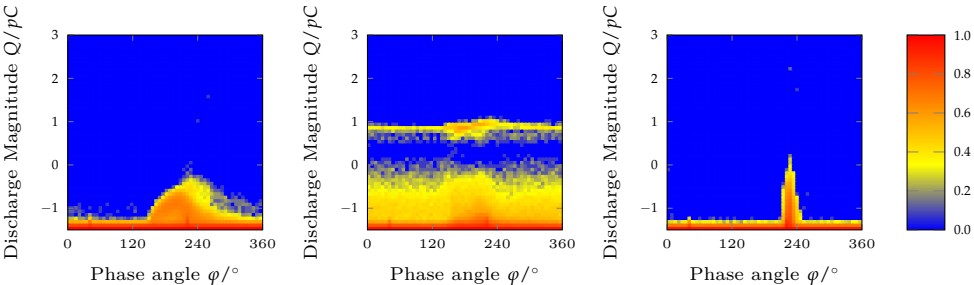

**Figure 3.** From left to right, examples of a hill, a plateau and a pylon. Many instances present a mix of these labels. The process for the preparation of the PRPD diagrams is described in the following sections.

**Table 1.** Number of files per combination of labels. This table is formatted like a truth table with ¬ standing for the logical negation. For example, there are 80 training instances with the label "Plateau & Pylon".

|  | **Plateau** | | **¬ Plateau** | |
| --- | --- | --- | --- | --- |
|  | **Pylon** | **¬ Pylon** | **Pylon** | **¬ Pylon** |
| Hill | 20 | 75 | 4 | 5 |
| ¬ Hill | 80 | 98 | 77 | 10 |

Other parasitic patterns are also present in some of the files. They are considered as noise and ignored in the subsequent classification task.

### 4.2. Creation of PRPD Diagrams

In the original data files, the amplitude of the discharges is signed, meaning that there are positive and negative discharges. However, in our case, keeping the signed amplitude to create PRPD diagrams is not relevant; indeed, it would lead to mirrored patterns, which do not help to solve the classification task, as well as a loss of density of the patterns. Thus, the absolute value is applied to the discharge amplitude column. The discharge amplitudes range between the measurement thresholds $10^{-2} pC$ and $10^{5} pC$ . The decimal logarithm is applied to map the discharge amplitudes to the interval $[-2, 5]$.

The raw data face challenges at both ends of the discharge amplitude range. On one hand, low-amplitude discharges consist mainly of noise. On the other hand, high-amplitude discharges are extremely scarce. Therefore, discharges with an amplitude in the interval $[-1.5, 3]$ are filtered out. The lower bound is empirically set to minimize the amount of noise. The upper bound is set to the 99.9th empirical percentile of the discharge amplitude distribution.

Without specification, it can be assumed that PRPD diagrams are generated based on the whole duration of the experiment. However, in some cases, only a temporal subset of the experiments is considered to generate PRPD diagrams. For example, in the evaluation section (Section 5), the impact of reducing the experiment length is evaluated by creating PRPD diagrams from the first seconds of the experiments. In this case, filtering is applied to the absolute time column, relative to the first timestamp. For example, if the time horizon is set to 10 s, then only rows with a timestamp belonging to the first 10 s of the experiment are kept.

Along the phase axis, the range $[0, 360]$ remains untouched.

It is now possible to generate the PRPD diagrams from the filtered data, as illustrated on Figure 4. Thus, the discharge amplitude and discharge phase are split in $n_\text{amp}$ and $n_\text{phase}$ bins, respectively. Then, the number of discharges falling into each square bin $(i, j)$ is calculated and used to define the raw intensity of the pixel $(i, j)$ of the PRPD diagram. Hence, the couple $(n_\text{amp}, n_\text{phase})$ defines the resolution of the generated PRPD diagrams. The higher the number of bins, the higher the detail of the PRPD diagram, as shown in Figure 5. It can be noted that the resolution of the images fed to the classification algorithms is much smaller than that of the original images shown in Figure 2. A specific evaluation is carried out to determine the optimal resolution in Section 5. Until then, that parameter is not set to a concrete value.

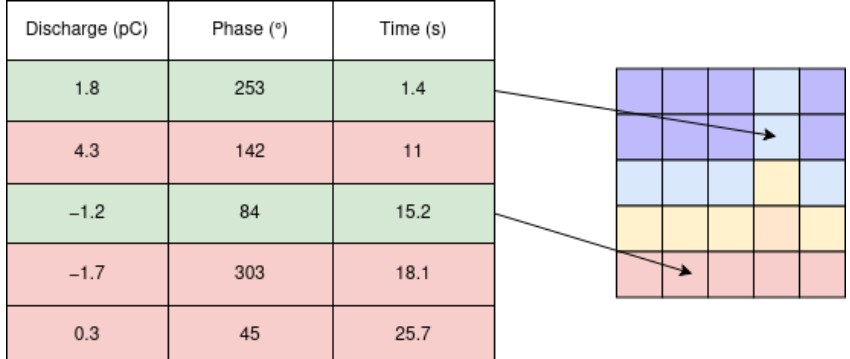

**Figure 4.** Illustration of the filtering and creation of a PRPD diagram. The table on the left is a subset of a PRPD file, which leads to the PRPD diagram represented on the right. Here, the discharge bounds are $[-1.5, 3]$, the time bounds are $[0, 25]$ and there are 5 bins along the discharge and the phase axis. The discharges are thus split into bins of $[-1.5, -0.6), [-0.6, 0.3), [0.3, 1.2), [1.2, 2.1), [2.1, 3]$ and used to build the rows, while the phase angles are binned into $[0, 72), [72, 144), [144, 216), [216, 288), [288, 360]$ and used to build the columns. After the application of the filtering, the red rows are dropped and only the green rows remain. Then, the green rows are added to the corresponding bins. For example, the lower green row is assigned to the second column and the first row bin.

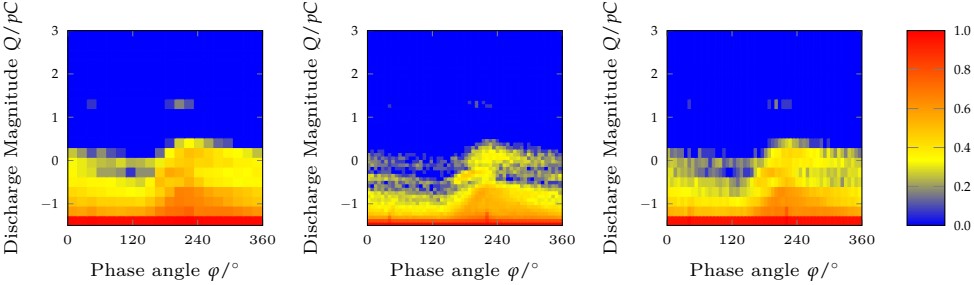

**Figure 5.** The same "Hill & Plateau" image under three different resolutions, namely $(20, 20)$, $(56, 56)$ and $(20, 56)$, from left to right. All the images have been logarithmically normalized.

The previous data filtering is not sufficient to balance the intensities of pixels. Indeed, pixels forming the patterns to be recognized have a typical intensity of 10, while a few noisy pixels remain, with an intensity greater than 700. Thus, a rescaling of the pixel intensities is necessary. Several methods have been considered, namely logarithmic normalization, power-law normalization, empirical cumulative distribution function and quantile transformation. A preliminary evaluation was conducted to select the logarithmic normalization. The logarithmic normalization of a PRPD diagram consists of two steps:

$$p_\text{log} = \log(1 + p_\text{raw})$$

$$p_\text{scaled} = \frac{p_\text{log} - \min(p_\text{log})}{\max(p_\text{log}) - \min(p_\text{log})} \tag{1}$$

where $p_{\mathrm{raw}}$ is the intensity of a pixel of the raw image and the extremal values are computed over the image. Given that $p_{\mathrm{raw}}$ is non-negative, the logarithmic transformation is always valid. Intuitively, the logarithmic normalization allows us to make the low-intensity pixels more visible relative to the high-intensity pixels. For example, taking one pixel of intensity $I_{\mathrm{low}} = 10$ and another of intensity $I_{\mathrm{high}} = 700$, the intensity ratio changes from $r_{\mathrm{raw}} = 70$ to $r_{\mathrm{scaled}} \approx 2.7$. Therefore, the normalization improves the contrast. Thanks to the min–max normalization, the pixel intensities lie in the interval $[0, 1]$.

### 4.3. Algorithms

The flow of the system is depicted in Figure 6. The preprocessing and picturization steps were presented in the previous sections, but we still need to specify the classifying machine learning algorithm, which takes images of fixed size with known labels as input and uses them for training a model. This model is a function that accepts unseen images and returns predictions on their labels. For this purpose, we utilize five different machine learning algorithms, which we introduce below. The learning of the model can be adjusted with so-called hyperparameters. Hyperparameters play a central role in nearly all machine learning algorithms and must be chosen by the researcher. The optimal choice of hyperparameters heavily depends on the problem at hand. Some challenges and objectives in the training of machine learning algorithms are outlined in Section 5. It is assumed that $N$ vectors $x_1, \ldots, x_N \in \mathbb{R}^d$ are observed, where each of the $d$ entries keeps the information on the gray-level of a specific pixel. The entries of the vectors are called features. The labels of the images are stored correspondingly in $y_1, \ldots, y_N$. For simplicity, it is assumed that $y_n \in \{-1, 1\}$, for $n = 1, \ldots, N$. Thus, every image belongs to exactly one of two classes. Except for convolutional neural networks (CNNs), the machine learning algorithms are combined with upstream dimension reduction techniques. However, to promote the accessibility of this section, it is assumed that the presented machine learning algorithms provide models that map images (and not their lower-dimensional projections) to labels. The utilized dimension reduction techniques are described at the end of this subsection.

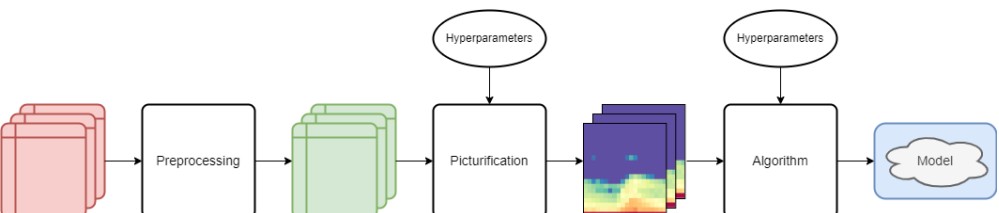

**Figure 6.** Data pipeline flow chart. The last two steps rely on hyperparameters that will be optimized.

#### 4.3.1. Classifiers

Logistic Regression

Under the logistic regression model, the probability of an image $x$ belonging to class 1 is given by

$$\mathbb{P}(Y = 1 | x) = \sigma(x^T \beta), \tag{2}$$

where $\beta \in \mathbb{R}^d$, and $\sigma$ is the sigmoid function. Intuitively, the logistic regression model classifies $x$ based on its orientation relative to the hyperplane of normal $\beta$. The sigmoid function $\sigma$ allows us to map the prediction to the interval $[0, 1]$. The logistic regression model is popular due to its simplicity, as it is fully characterized by $\beta$. The impact of each pixel in $x_j$ on the prediction is given by the corresponding parameter $\beta_j$. The larger $\beta_j$, the stronger the pixel intensity $x_j$, forcing the prediction towards class 1. In order to avoid overfitting for high-dimensional data as images, we search for $\beta$ by minimizing

$$\hat{\beta} = \arg\min_{\beta \in \mathbb{R}^d} -L(\beta | x_1, \ldots, x_N) + \lambda h(\beta), \tag{3}$$

where $h(\beta)$ penalizes complex models (for example, $\beta$ with many non-zero entries) and $L$ is the log-likelihood function of the logistic regression model. As hyperparameters, one has to choose the scalar value $\lambda$ and the function $h$. In the experiments below, $h$ is chosen to be the sum of the absolute values of $\beta$ (i.e., its $l^1$ norm) and $\lambda$ is varied. An increasing $\lambda$ prefers models of lower complexity, while $\lambda = 0$ corresponds to no penalization of complexity. If there are more than two classes an image $x$ can belong to, then we use a classifier chain, where we train a different model for every label and chain these together [19].

Support Vector Machine

Assuming that the images $x_1, \ldots, x_N$ are ordered in such a way that $x_1, \ldots, x_r$ belong to class 1 and $x_{r+1}, \ldots, x_N$ belong to class $-1$, then support vector machines (SVMs) search for a hyperplane that splits the data points $x_1, \ldots, x_r$ and $x_{r+1}, \ldots, x_N$ across different sides. Furthermore, SVMs try to maximize the minimal distance of any data point to the hyperplane. A two-dimensional example, for which such a hyperplane exists, is depicted in Figure 7. As data are seldom linearly separable (i.e., such a separating hyperplane exists), one can map the data into a higher-dimensional feature space. If chosen correspondingly, distances and angles in this feature space can be calculated via a so-called kernel. As hyperparameters, we need to choose the mapping (i.e., the kernel) and a regularization factor, which ensures that hyperplanes of lower complexity are preferred. Furthermore, we choose the radial basis function as kernel, which offers another free parameter. An introduction to SVMs can be found in [20].

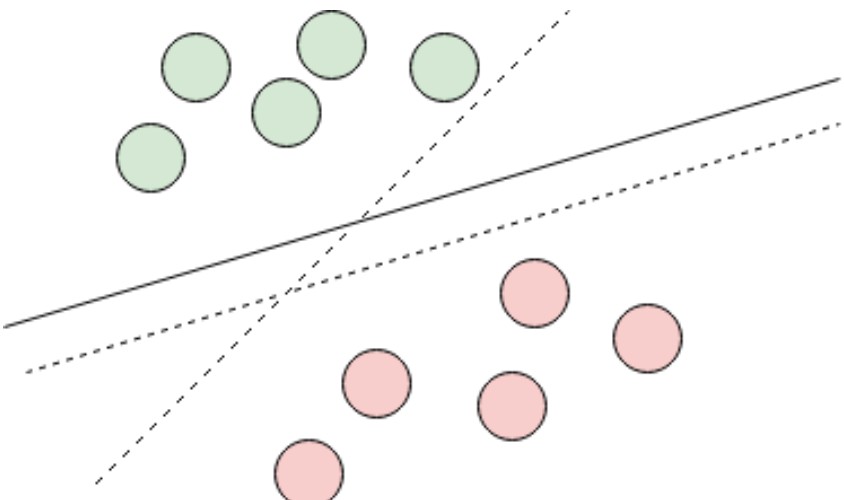

**Figure 7.** In the linearly separable scenario above, the support vector machine (SVM) finds the hyperplane (dotted and solid lines), which has the maximal distance to the nearest data point. While all drawn hyperplanes separate the data, the solid line has the largest distance to the closest data point and is thus the result of the SVM algorithm. An unseen data point is assigned to the green class if it falls above the solid line and to the red class if it falls below the solid line.

Random Forest

A decision tree can be constructed by a root node, which contains all data points. Then, using the features of $x_1, \ldots, x_N$ it splits the data into two child nodes. Applying this pattern recursively on the child nodes, we obtain a decision tree. A concrete example of a decision tree is shown on Figure 8. A new data point $x$ to classify is assigned to a label by walking through the decision tree until a node with no children is reached. We call nodes with no children leaf nodes. By convention, one assigns the new data point $x$ to the label that builds the majority in the respective leaf node. During the training of the decision tree, the algorithm identifies at each node the criterion of the features that separates instances with different labels $y$ as well as possible. Decision trees are very simple but often do not provide enough complexity. However, they are popular in ensemble methods, two of which are outlined below. In the case of more than two classes (multiclass) or if an instance can even

belong to an arbitrary number of labels (multilabel), the learning of the decision tree can be adjusted [21].

One example of an ensemble method is the random forest (RF) [22]. During the training of an RF, the algorithm trains multiple decision trees, while randomly subsampling the images and features. To predict the labels of an image *x*, every decision tree makes a prediction. The prediction of the RF is then the majority vote of the decision trees.

There are numerous hyperparameters for an RF. However, we vary only the number of decision trees and their maximal depth, which is the maximal number of junctions one needs to pass to arrive at a leaf node. A lower number results in simpler models.

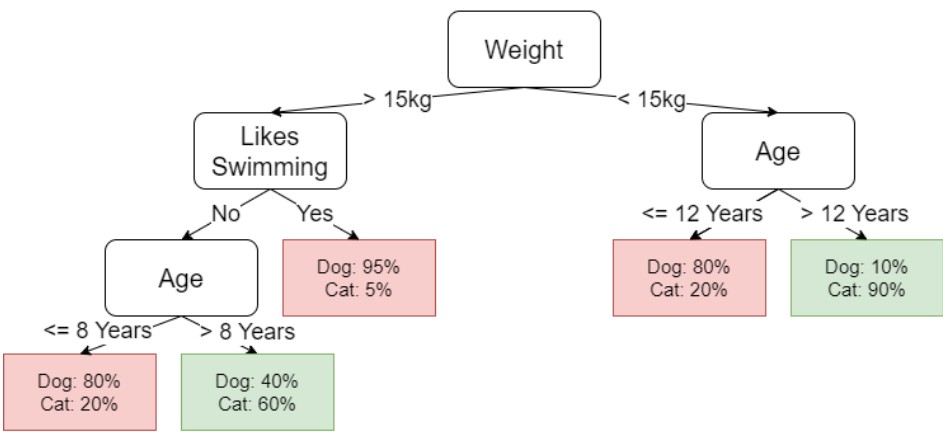

**Figure 8.** A decision tree to distinguish a cat from a dog based on the features Weight, Age, and Likes Swimming. The leaf nodes are colored in green and red depending on the prediction. An example animal, which weighs 18 kg, does not like swimming and is 4 years old, is predicted to be a dog. The depth of the depicted decision tree is three.

XGBoost

XGBoost also uses a combination of decision trees. However, the training algorithm to derive the decision trees is different. Indeed, the trees are derived sequentially to steadily improve the efficiency of the combination of trees. Given a certain loss function *L* and a learning rate $\alpha$, the algorithm finds a first decision tree $D^1$ that approximately minimizes

$$L(D) = \sum_{n=1}^{N} L(y_n, D(x_n)) + h(D), \tag{4}$$

where *h* measures the complexity of a tree *D* and prevents excessive overfitting. For this decision tree $D^1$, we denote the class prediction of $x_n$ by $\widehat{y_n}^1 = \alpha D^1(x_n)$. Then, XGBoost iteratively searches at step $t+1$ for a decision tree $D^{t+1}$ that approximately minimizes

$$\sum_{n=1}^{N} L(y_n, \widehat{y_n}^t + D(x_n)) + h(D). \tag{5}$$

Finally, the prediction for $x_n$ is given by

$$\widehat{y_n}^{t+1} = \sum_{v=1}^{t+1} \alpha D^v(x_n). \tag{6}$$

Several hyperparameters allow us to tune the XGBoost method. In the experiments, we vary the learning rate $\alpha$, which determines the magnitude of the contribution of each tree, and the maximal depth of the decision trees. An extensive introduction to XGBoost is provided in [23].

Convolutional Neural Networks

Artificial neural networks are machine learning methods inspired by the organization of human neurons. A neural network consists of a directed graph of parameterized nodes organized in layers, which can be trained to solve classification tasks inter alia. Convolutional neural networks (CNN) are particularly suitable for image classification [24]. The specificity of CNNs lies in their convolutional layers. Those layers are made of learned kernels, which are convolved along the input images to generate activation maps. These activation maps capture meaningful characteristics of the image and facilitate the classification made by the dense layers of the network. Thus, as opposed to the previously presented algorithms, no dimension reduction technique is needed. Raw images can be directly fed to the network.

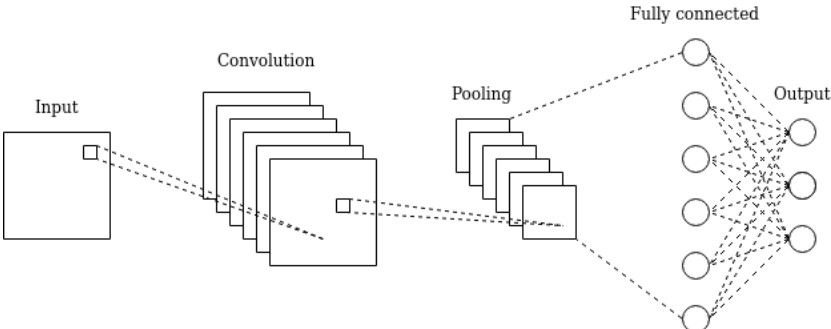

**Figure 9.** The architecture of the CNN model. The convolution and pooling layers are in charge of the feature selection, while the classification is made by fully connected layers.

In this work, the considered CNN architectures have only one convolutional layer and two dense layers, as shown on Figure 9. This choice is justified by the limited size of the dataset, which prevents the efficient training of deeper networks. The activation functions are rectified linear units, except for the output layer, which uses a sigmoid function. This allows us to output a score ranging between 0 and 1 for each of the base classes. $L_2$ regularization is applied to the weights of the network to prevent overfitting. At training time, the Adam optimizer [25] is used to fit the weights of the network.

4.3.2. Dimension Reduction Techniques

Principal Component Analysis

For simplicity, let us assume that the data are centered, i.e., that $\sum_{n=1}^{N} x_n = 0 \in \mathbb{R}^d$ (this can be obtained by subtracting the mean from each data point). The principal component analysis (PCA) [26] searches for the linear combination of the $d$ features that contains the largest variation. More precisely, the first principal component $a_1 \in \mathbb{R}^d$ is

$$a_1 = \arg\max_{a \in \mathbb{R}^d} \sum_{n=1}^{N} (a^T x_n)^2. \tag{7}$$

Geometrically, $a_1$ is the normal of the hyperplane that separates the centered point cloud $(x_n)_{n=1,\dots,N}$ best. For $k > 1$, the $k$th component $a_k \in \mathbb{R}^d$ is the vector maximizing Equation (7) while being orthogonal to the subspace spanned by $\{a_1, \dots, a_{k-1}\}$. The dimensions of the data $x_1, \dots, x_N$ can then be reduced by $z_n = (a_1^T x_n, \dots, a_k^T x_n) \in \mathbb{R}^k$, $n = 1, \dots, N$ and $k \leq d$. Typically, $k$ is smaller than $d$ by several orders of magnitude to allow high compression.

Non-Negative Matrix Factorization

Let $\mathbf{X} = (x_1, \dots, x_N)^T \in \mathbb{R}^{N \times d}$ be the data matrix with non-negative entries. For a given $k \leq d$, non-negative matrix factorization (NMF) searches for the non-negative matrices $H \in \mathbb{R}^{N \times k}$ and $W \in \mathbb{R}^{k \times d}$ such that

$$||\mathbf{X} - HW||_F \tag{8}$$

is minimized, where $|| \cdot ||_F$ is the Frobenius norm on matrices. Intuitively, the columns of $W$ can be interpreted as basis vectors of the original space. The coefficients of $H$ show how to combine these basis vectors in order to reconstruct the original data. The dimensions of the data can then be reduced by considering $h_n \in \mathbb{R}^k$ instead of $x_n$, and $h_n$ is the $n$th row of the matrix $H$. NMF is often able to learn good lower-dimensional representations for images [27].

*4.4. Evaluation Methodology*

4.4.1. Classification Metrics

Section 4.1 shows that a single PRPD diagram can belong to multiple classes. Therefore, classifying PRPD diagrams is a multi-label classification problem. It implies that the labels $l_n$, with $n \in \{1, \dots, N\}$, can be encoded as boolean vectors of a size equal to the number of classes $C$. The order of the coordinates of the label vectors follows that of the classes "Hill", "Plateau" and "Pylon". For example, assuming the first file is labeled as "Hill & Pylon", then $l_1 = [1, 0, 1]$. In the following, $\hat{l}_n$ denotes a prediction made for the $n$-th file.

To evaluate the ability of the models to solve the classification task, we consider the Hamming loss, which is the fraction of misclassified individual labels. Formally, it is defined by

$$\text{hamming loss} = \frac{1}{NC} \sum_{n=1}^{N} \sum_{c=1}^{C} \text{xor}(l_{n,c}, \hat{l}_{n,c}). \tag{9}$$

The Hamming loss is chosen as the main decision metric. This is reasonable as

$$\frac{1}{NC} \sum_{n=1}^{N} \sum_{c=1}^{C} l_{n,c} \tag{10}$$

is well between 0 and 1.

4.4.2. Nested Cross-Validation Methodology

The dataset at our disposal contains fewer than 1000 files. It can thus be considered as small. Therefore, special emphasis must be placed on the algorithm and model selection methods to ensure that the selected model generalizes well on unseen data. In this work, the nested cross-validation methodology is chosen to tune and compare the classification methods [28,29].

Nested cross-validation is a generalization of cross-validation, which is a resampling method that allows the training and testing of a classification method over rotating splits of the dataset [30], as shown on Figure 10. By averaging the classification scores obtained over the different splits, it is possible to compute the value of a fitted model derived from a particular hyperparameter configuration. However, it is not possible to optimize the hyperparameter configuration based solely on the cross-validation score. Indeed, this would lead to the overfitting of the hyperparameters. Nested cross-validation addresses that issue by nesting two cross-validation loops: the optimal hyperparameter configuration is found in the inner loop and then the ability of the tuned algorithms to generalize to unseen data is evaluated in the outer loop.

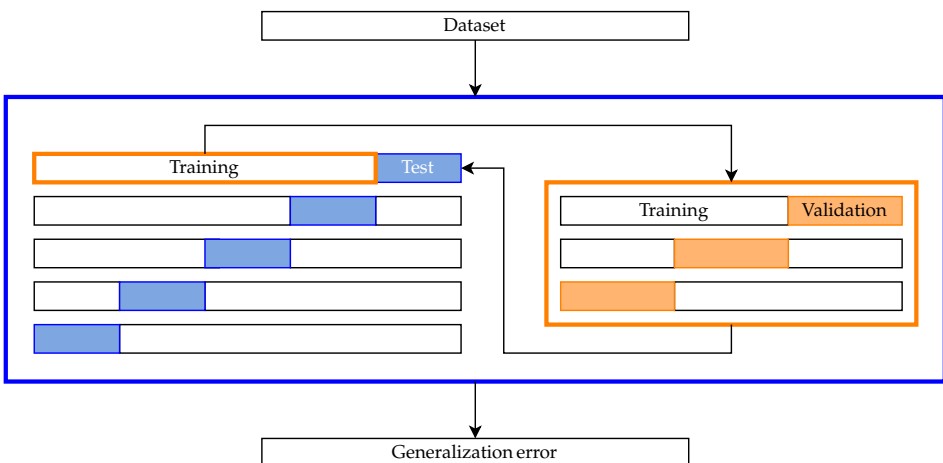

**Figure 10.** Illustration of the nested cross-validation. In this example, the inner and outer cross-validations are represented in orange and blue and have 3 and 5 folds, respectively.

All that remains is to set the number of folds both for the inner and outer cross-validation. On one hand, the number of folds must be restricted to preserve the computational feasibility of the selection process. On the other hand, shrinking the number of folds negatively impacts the statistical significance of the generalization score. A trade-off is found by setting the number of folds of the inner cross-validation to 5 and the number of folds of the outer cross-validation to 10.

## 5. Results

In this section, we evaluate the different machine learning algorithms using the methodology described in the previous section. As the resolution of the PRPD diagrams is somewhat arbitrary, we start by evaluating how different algorithms perform with images of different resolutions (Section 5.1). In the subsequent subsection, we use the obtained optimal combination of algorithm and resolution, where we evaluate the performance of the algorithms for different durations of the PRPD recordings (Section 5.2). This is of particular interest as it hints at the applicability of the DC partial discharge method for industrial routine tests. We present which labels are hard to identify (Section 5.3) before we apply methods that aim to explain the decision of a machine learning model in Section 5.4. In such a way, structures learned by the algorithm can be visualized and compared with the expert assessment.

### 5.1. Choosing the Image Resolution

For each combination of algorithm and PRPD diagram shape, a nested cross-validation is executed to determine the ability of the given algorithm to generalize on pictures of that given size. We vary the hyperparameters of the algorithms within the inner loop. For details of the hyperparameter search spaces, we refer the reader to Appendix A. The results of the experiments are presented in Table 2.

The search for the optimal shape does not lead to a unified result. This is probably due to the low sample size and a few hard-to-classify samples. Nevertheless, the algorithms logistic regression, support vector machine, random forest, and XGBoost prefer shapes, where at least one side is not too large. At the same time, shapes with a small number of pixels show a bad performance. Intuitively, the algorithms need enough information to learn the correct patterns. At the same time, too many pixels lead to variance that cannot be eliminated sufficiently through dimension reduction. A heatmap of the results for RFs can be found in Figure 11.

**Table 2.** Image resolution experiment results for the different considered algorithms. The mean of the Hamming loss was calculated on the hold-out set in the outer loop of the nested cross-validation. The preferred hyperparameters were found using a regular cross-validation on the optimal shape.

| Algorithm | Optimal Shape | Mean (Hamming Loss) in Outer CV | Preferred Hyperparameters (on Separate CV) |
|---|---|---|---|
| CNN | $74 \times 38$ | 0.107 | Batch Size: 8<br>Dense Layer Width: 128<br>Kernel Size: 4<br>Learning Rate: $10^{-5}$<br>Number of Filters: 48 |
| Logistic Regression | $20 \times 20$ | 0.118 | Dimension Reduction: NMF<br>Regularization: 100<br>Dimensions Projection: 37 |
| Support Vector Machine | $20 \times 38$ | 0.128 | Dimension Reduction: NMF<br>Regularization: 10<br>Kernel Coefficient: 10<br>Dimensions Projection: 25 |
| Random Forest | $92 \times 20$ | 0.106 | Dimension Reduction: NMF<br>Maximal Depth Trees: 20<br>Dimensions Projection: 45 |
| XGBoost | $20 \times 74$ | 0.106 | Dimension Reduction: NMF<br>Maximal Depth Trees: 11<br>Dimensions Projection: 45<br>Learning Rate: 0.3 |

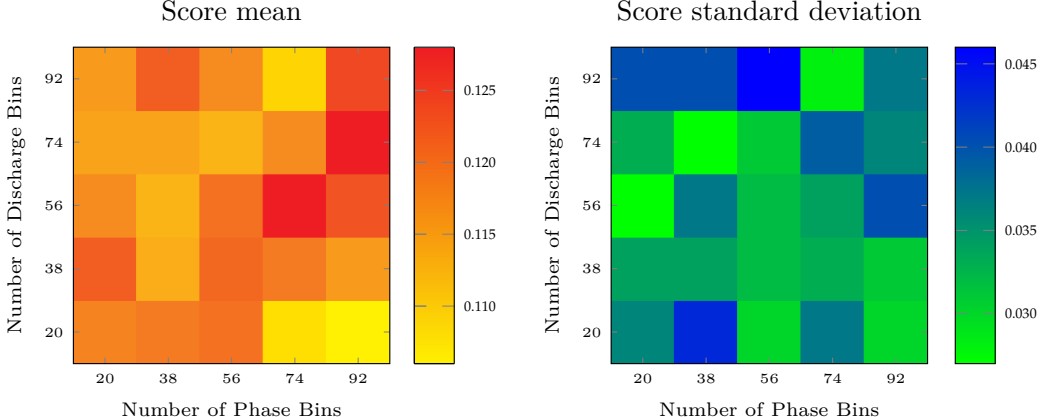

**Figure 11.** Generalization scores obtained after nested cross validation of the Random Forest (RF) algorithm for multiple image resolutions. For each resolution, the mean and the standard deviations of the outer fold scores are displayed on the left and right heatmaps, respectively. The optimal resolution for this algorithm (RF) is $92 \times 20$.

On the contrary, the results of the CNNs seem to be robust with respect to the resolution. Figure A1 in the Appendix A shows several local optima without a clear trend. Still, resolutions where the number of phase bins is significantly greater than the number of discharge bins under-perform compared to the other resolutions. Again, the high variance of classification scores on the outer folds can be explained by the presence of a few hard-to-classify samples. In the following, the algorithm and shape with the best mean score are chosen: the random forest with a resolution of $92 \times 20$. Despite the small sample size compared to typical deep learning applications, CNNs are competitive with the best-performing algorithms. For experiments with a larger sample size, it is thus absolutely reasonable to consider CNNs, and their performance might even be superior.

### 5.2. Short-Term Identification Ability of PRPD Patterns

Here, the impact of time spans is evaluated. In a general setting, we could repeat the experiment of Section 5.1 for every time horizon. However, considering the stationarity of the process, it is postulated that the best coupling of algorithm and image resolution for the complete time span is also near-optimal for shorter time intervals. Thus, the random forest with shape $92 \times 20$ from Section 5.1 is selected. Its performance is assessed using pictures with time spans of 1.0, 1.5, 5.0, 10.0, and 30.0 s. For every time interval, a nested cross-validation is run to estimate the expected generalization ability of the algorithm, as described in Section 4.4. The hyperparameter search space is chosen as in Section 5.1. The outcome of that evaluation is shown in Figure 12, where we report the mean of the Hamming loss over the ten outer cross-validation folds (blue line) and the standard deviation (blue area). We see that the Hamming loss drops within the first two seconds, before it starts to flatten out.

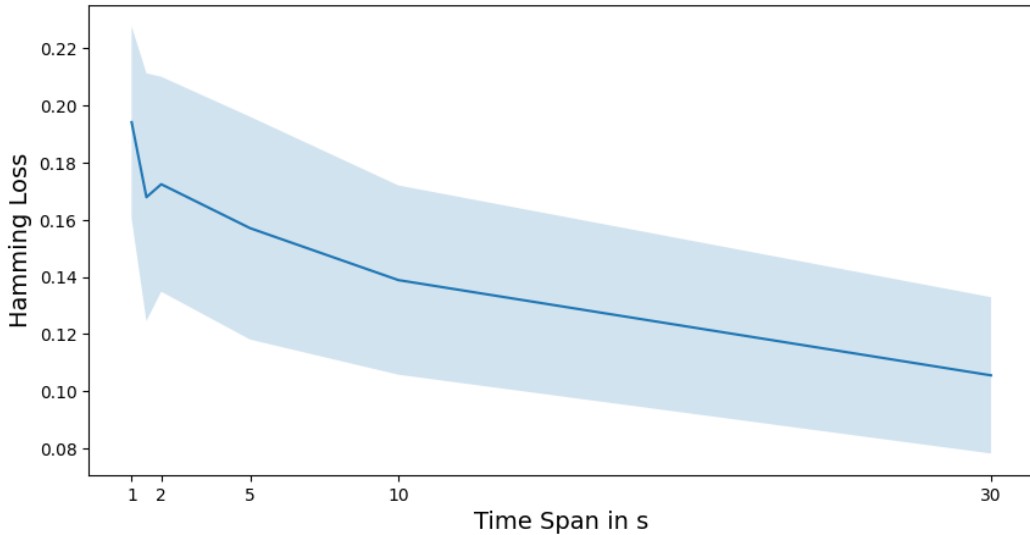

**Figure 12.** Nested cross-validation scores (solid line) and standard deviation (shaded area) for the random forests with resolution $92 \times 20$ and time spans of $\{1.0, 1.5, 5.0, 10.0, 30.0\}$.

This is in line with our visual analysis of the diagrams. For many instances, stationarity is achieved early. That means if we cut out arbitrary time spans (e.g., between 5 and 10 s), the discharge diagrams look very similar for one specific test object. Thus, the pattern can already be detected within the first seconds of the procedure.

### 5.3. Performance by Label

Again, we use the RF algorithm with shape $92 \times 20$. Additionally, we take the optimal hyperparameters learned through cross-validation as presented in Table 2. Another five-fold cross-validation is run. As usual, four folds are used for training the model. The predictions of the model of the test-fold diagrams are compared with their true labels. As every image appears in the test-fold exactly once, the numbers are summed up by label. In total, there are 369 images, and the number of false positives is reasonably low , as shown in Table 3. Unfortunately, the number of false negatives is relatively high. For a classification procedure in series production, the main goal is to reduce the number of false positives as much as possible, as they pose a threat to the final product quality. At the same time, the number of false negatives should be reasonably small in order to avoid having to sort too many pieces and thus increase the scrap rate. The calibration of the model is beyond the scope of this paper. In fact, we implicitly consider false positives to be as harmful as false negatives during our analyses.

**Table 3.** The predictions of the model on unseen images by label. We see that the number of diagrams and the number of false positives are correlated, as are the number of diagrams without patterns and the number of false negatives. The predictions for the label "Plateau" are better than for the labels "Pylon" and "Hill". The latter are harder to distinguish with the human eye.

| Label | Diagrams with Pattern | Diagrams without Pattern | False Positive | False Negative |
|---|---|---|---|---|
| Plateau | 273 | 96 | 18 | 12 |
| Pylon | 181 | 188 | 18 | 25 |
| Hill | 105 | 264 | 10 | 45 |

In the second step, we combine the labels "Plateau" and "Hill" as "faulty". That means as soon as one label is either "Plateau" or "Hill", the image is considered as a "Fail". In any other case, the image is considered as a "Pass". Using the results of the cross-validation above, we observe the confusion matrix of Figure 13.

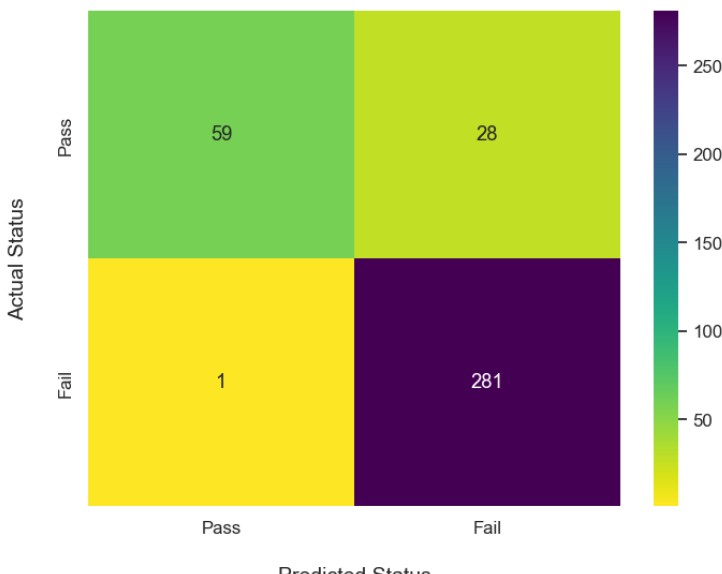

**Figure 13.** Confusion matrix, where labels are grouped as "Fail" and "Pass". We consider an image as a "Fail" if a diagram has at least one of the labels "Hill" or "Plateau". If the diagram has neither of these labels, we consider it as a "Pass". We see that the model predicts only one false negative. However, there are also 28 false positives. As false negatives are the most harmful in a series production, we aim for an over-sensitive system.

Due to the higher share of "Fail" images, the system detects images with insulation faults very well. The number of false negatives is reasonable.

### 5.4. Explainability

The focus of this work is to show that the root causes behind PRPD images can be automatically detected, and this can be achieved in a reasonable time span. In many domains, models which are hard for humans to explain have shown a superior performance for predictions. These models are called black box models. All presented algorithms but the logistic regression can be understood as black box models. While the lack of interpretability is unfortunate, applications focusing on predictions for unseen data often accept no intrinsic explainability. However, we would like to ensure that the model performs well on unseen images or on perturbations of known images [31]. This is particularly relevant for the rather low number of images in our dataset. In recent years, the interpretation of machine learning models [32,33] has become a growing research topic. These methods are summarized by the

term explainable AI (xAI). Among other applications, xAI can be applied to detect possible vulnerabilities and weaknesses and to uncover possible improvements in the model [34]. As domain experts can formulate why an image belongs to a certain label, the explanations of the ML method are compared with the expert assessment. This serves two purposes:

1. In cases where the explanations of the model match those of the expert, one can be confident that the algorithm has learned the correct patterns and the model will also apply them to unseen images.
2. It is important to rule out that the algorithm has not learned specific artifacts in the training images, which have a high predictive power but contain no information for unseen data [35].

Both aspects hint at the robustness of the model and the ability to transfer the learned patterns to unseen images.

### 5.4.1. Local Interpretable Model-Agnostic Explanations (LIME)

Local Interpretable Model-Agnostic Explanations (LIME) is an algorithm that can explain the predictions of a classifier by approximating it locally with an interpretable model [32]. For example, applying the LIME method to a classifier and a particular PRPD allows us to identify the areas of the PRPD which are deemed meaningful by the classifier to determine the presence of "Hill", "Plateau" and "Pylon". LIME is particularly suitable to our study case compared to other xAI methods like SHapley Additive exPlanations (SHAP) [36]. Indeed, the heuristic approach of LIME drastically reduces the computational cost of an explanation, which matters when dealing with highly dimensional data like PRPDs.

### 5.4.2. Interpretation of the Classifier

Again, we consider the best-performing algorithm of Section 5.1, which is the random forest. However, to improve the presentation, we do not consider the resolution $92 \times 20$ but $38 \times 54$, which also provided a reasonable result (mean of Hamming score in outer loop is 0.112) but is better for visualization purposes. Again, we could consider different time spans. However, due to the stationarity of the process, we focused on the PRPDs containing all recorded discharges. In the beginning, we ran a cross-validation in order to determine suitable hyperparameters for the random forest. Then, one specific example was chosen for examination. The remaining images and the identified hyperparameters were used for training the model. Afterwards, LIME provided explanations for the image under inspection. It is emphasized that this image was not used for training.

This subsection is of an exemplary nature. However, two representative instances are shown.

### 5.4.3. "Pylon" Predicted, True Label Is "Pylon"

We examine a PRPD image, whose true label is "Pylon", which was also correctly predicted by the model. We analyze the prediction with respect to the labels "Hill" and "Pylon". We see in the left canvas of Figure 14, that the model considers the red shaded areas to not be compatible with the label "Hill". These red areas outweigh the green areas, which support the label prediction "Hill". For the true label "Pylon", the green areas in the middle canvas show that the sharp spike and the narrow area of discharges are typical for the label "Pylon" and there are no particular regions which contradict this. The right canvas shows the true PRPD image.

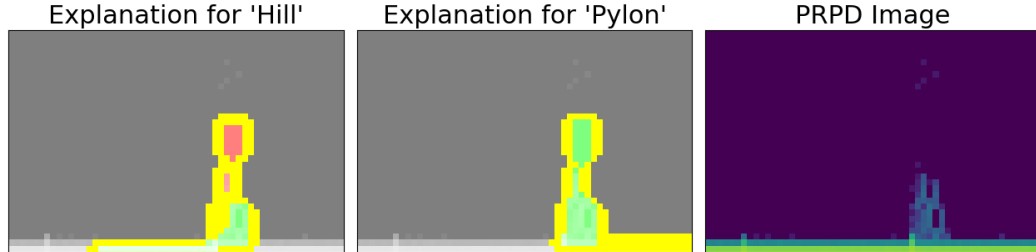

**Figure 14.** This image was labeled "Pylon", which was also correctly classified by the model. The areas covered in green show the areas which led to the correct prediction. The explanation matches the expert assessment, as "Pylon" is characterized by a narrow spike.

5.4.4. "Plateau" Predicted, True Labels Are "Pylon" and "Plateau"

In the second example, we depict in Figure 15 a hard-to-classify image, whose true labels are "Pylon" and "Plateau". The model correctly predicts the label "Plateau", but misses the label "Pylon". The red areas in the left canvas indicate which parts of the image lead to a lower prediction score for the label "Pylon" according to the model. We see that the discharges with a lower phase angle contradict the model's concept of the label "Pylon". This is due to the fact that "Pylons" in other diagrams are typically characterized by a narrow spike of discharges at a higher phase angle. At the same time, the discharges along the phase angle are identified as driving factors for the prediction of "Plateau", which matches the expert assessment.

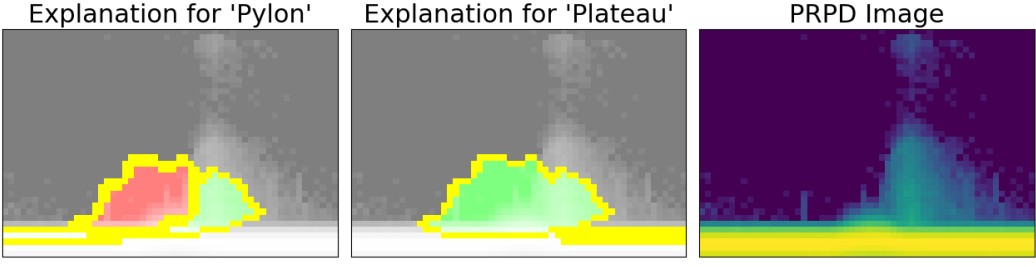

**Figure 15.** This image was assigned the labels "Pylon" and "Plateau". While "Pylon" was not detected, the driving areas are colored in red in the left canvas. The green areas support the decision for the label "Pylon" but are outweighed by the red areas. The middle canvas shows the driving areas for the correct assignment of the label "Plateau". There are no areas contradicting this prognosis. The right canvas shows the original PRPD image.

## 6. Discussion and Conclusions

In this paper, we have presented an automated pattern recognition system, which is capable of detecting faulty insulation in the production process of traction battery systems. Defects like solid impurities or voids can occur during the insulation process, e.g., due to missing technical cleanliness. Based on DC partial discharge diagnostic diagrams, we applied computer vision methods. Contrary to plain-vanilla computer vision use cases, the creation of the images (the diagrams) is part of the presented pattern recognition system. We have shown that the chosen resolution has a smaller impact on the effectiveness of the image classification algorithms, while the pixel intensity had a large influence. Comparing different classification algorithms, we have shown that the system is able to identify faulty pieces and their root causes. Although our database consists of only 369 images, convolutional neural networks known to be effective for classifying real-world images have shown a competitive performance compared to other methods based on trees and bagging or boosting, such as random forests and gradient boosting.

The common misconception that the application of DC partial discharge diagnostics is always time-consuming [4] can be refuted in this application. Thus, the presented method is competitive with leakage current measurements [5]. This is demonstrated in

Figure 12, which shows a strong classification quality even for short testing intervals (e.g., 2 s). However, as the loss of accuracy within the first few seconds is based on physical phenomena like ignition delay and the varying avalanche behavior of discharges [5], some patterns appear later. Hence, one must balance the system's precision and the testing duration. Figure 12 serves as a basis for this decision.

Aside from routine tests, the DC partial discharge diagnostic can also be applied as an initial test for new insulation materials and geometries. In this case, the test duration should be increased to ensure the highest accuracy. In contrast to the existing literature [4], our results show that this can be achieved within minutes rather than hours.

In this paper, the algorithms are designed to separate specific discharge patterns, which can be used for the identification of root causes. However, the discharge patterns also fall into the two categories "Pass" and "Fail". Reassigning the labels accordingly, we have shown that the system rarely misses defects, i.e., the number of false negatives is very low. At the same time, the number of falsely identified defects, i.e., false positives, is reasonable. This indicates that faulty pieces are detected reliably.

In the manufacturing of electric vehicles, insulation tests are applied frequently to ensure the highest quality standards. Repeated testing increases the total testing time. Thus, the manufacturer can guarantee that no mechanical, thermal, or electrical stress leads to defects during production. However, the faults should be detected as early as possible. Thus, resources can be saved by, for example, recoating the batteries before they are non-destructively assembled in modules and packs [37]. In this context, introducing DC partial discharge diagnostics and pattern recognition in the automotive sector leads to the gain of valuable information on the insulation quality in traction battery systems. Hence, DC partial discharge diagnostics contribute to the sustainable development of BEVs by increasing the resource efficiency and thus by reducing carbon emissions in the transportation sector [1].

For future work, improvements in the classification algorithm are to be considered. In particular, one idea would be to replace the PRPD encoding, which was derived thanks to expert knowledge of the field, with a learned encoding. For example, a transformer neural network [38] could be trained on the temporal series of discharges and afterwards be applied to embed those series in a vector space of reduced dimensions. Such an extension would allow us to compare the domain-aware algorithm presented before with a purely data-driven method.

Additionally, the calibration of the system toward minimizing false negatives is of interest. This results in an improved pass–fail test.

**Author Contributions:** Conceptualization, I.F. and M.K.; methodology, I.F., M.K. and R.S.; software, M.K. and R.S.; validation, I.F., M.K. and R.S.; investigation, I.F., M.K. and R.S.; resources, I.F. and M.K.; data curation, I.F.; writing—original draft preparation, I.F., M.K. and R.S.; writing—review and editing, I.F., M.K. and R.S.; visualization, I.F., M.K. and R.S.; supervision, I.F. and M.K.; project administration, I.F. and M.K.; funding acquisition, I.F. and M.K. All authors have read and agreed to the published version of the manuscript.

**Funding:** This research was funded by BMW AG.

**Data Availability Statement:** Due to commercial restrictions, supporting data is not available.

**Conflicts of Interest:** The authors declare no conflict of interest.

## Abbreviations

The following abbreviations are used in this manuscript:

| | |
|---|---|
| BEV | Battery electric vehicles |
| CNN | Convolutional neural network |
| $CO_2$ | Carbon dioxide |
| LIME | Local interpretable model-agnostic explanations |

| | |
|---|---|
| NMF | Non-negative matrix factorization |
| PCA | Principal component analysis |
| PD | Partial discharge |
| PoC | Proof of concept |
| PRPD | Phase-resolved partial discharge |
| RF | Random forest |
| SHAP | Shapley additive explanations |
| SVM | Support vector machine |
| TO | Test object |
| xAI | Explainable artificial intelligence |

## Appendix A. Hyperparameter Search Spaces

**Table A1.** Hyperparameter Search Spaces for the presented algorithms. For all algorithms and all presented experiments we conducted a grid search, i.e., we evaluated all possible combination of the hyperparameters.

| Algorithm | Hyperparameter Search Space |
|:---:|:---:|
| CNN | Batch Size: $\{4, 16\}$<br>Dense Layer Width: $\{64, 192\}$<br>Kernel Size: $\{2, 8\}$<br>Learning Rate: $\{10^{-5.5}, 10^{-4}\}$<br>Number of Filters $\{32, 96\}$ |
| Logistic Regression | Regularizer: $\{1, 10, 100\}$<br>Dimensions Projection: $\{15, 22, 30, 37, 45\}$ |
| Support Vector Machine | Regularizer: $\{0.1, 1, 10\}$<br>Parameter Kernel: $\{0.1, 1, 10\}$<br>Dimensions Projection: $\{15, 25, 35\}$ |
| Random Forest | Maximal Depth Trees: $\{15, 20\}$<br>Dimensions Projection: $\{20, 25, 30, 35, 40, 45, 50\}$ |
| XGBoost | Maximal Depth Trees: $\{6, 11\}$<br>Dimensions Projection: $\{25, 30, 35, 40, 45, 50\}$<br>Learning Rate: $\{0.3, 0.5\}$ |

**Figure A1.** Generalization scores obtained after nested cross validation of the Convolutional Neural Network (CNN) algorithm for multiple image resolutions. For each resolution, the mean and the standard deviations of the outer fold scores are displayed on the left and right heatmaps, respectively. The optimal resolution for CNN is $38 \times 74$.

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
