# Peer review of "Computer Vision for DC Partial Discharge Diagnostics in Traction Battery Systems"

_wevj, doi:10.3390/wevj14080222_

Round 1

Reviewer 1 Report

Lines 55 to 59: Do not use first person singular.

Lines 108 to 111: Do not use first person singular.

Lines 113 to 114: Do not use first person singular.

Lines 121 to 126: Do not use first person singular.

Lines 200 to 203: Do not use first person singular.

Lines 209 to 210: Do not use first person singular.

Lines 250: Do not use first person singular.

Lines 378 to 381: Do not use first person singular.

Lines 387 to 388: Do not use first person singular.

Lines 435 to 440: Do not use first person singular.

The lines from 473 to 488 are written in very colloquial language. This section should be revised.

Reviewer 2 Report

-All equations must be numbered.

-Write assumptions as a bullet form.

-Write the logic behind the each equation.

-Write your motivation clearly. what is the difference between this study and other works? Provide a separate section.

-What things should be done in future? provide a separate section.

Reviewer 3 Report

The article called "Computer Vision for DC Partial Discharge Diagnostics in Traction Battery Systems" is interesting but needs some clarification.

I write a list of changes/clarifications, not indexed by importance  but  found by reading the article.

1) in the introduction I don't like the selected articles. A review of a conference is used to define partial discharges, when there are much more important articles than books. Reference is made to a doctoral thesis when there are journal or conference articles of the same author

2) For the scheme in figure 1, I would like the TO to be defined. This is an important point. The ac-side system is an external part of the EV, so it must be assumed that it is in the dc part of the EV.

3) Figure 2 is interesting. It is necessary to enter the types of patterns on which article they are defined (or put it in the text) but especially the observation time period (the degrees refer to a sinusoid with a period). Is the variable trend in the upper part of the graphs due to the ripple of the rectifier (in electric vehicles there shouldn't be any) or to the superimposed sine wave?

4) is the dataset available? Does it have its own internet address from which it can be downloaded?

5) better describe figure 5, there are no units of measurement (pC, °, s) and how they are boxed to create the pattern of the second part of the figure

6) the method part has a great development, it could be shortened

7) part 4 is well developed

8) In the discussion and conclusion reference is made to Battery electric traction, I would like to see the case in which the defects discussed above occur

Round 2

Reviewer 3 Report

the article has been improved as requested, cleared for publication